# OpenReview forum: "Stop Summation: Min-Form Credit Assignment Is All Process Reward Model Needs for Reasoning"
_NeurIPS.cc/2025/Conference — NeurIPS 2025 poster_

### Official Review · Reviewer_ydx9 · 2025-06-02

**Clarity:** 3
**Significance:** 2
**Originality:** 2
**Rating:** 4
**Confidence:** 3

**Summary:**

This paper introduces a novel framework named PURE, which uses a min-form credit assignment to formulate the value function as the minimum of future rewards. The value function is thus also constrained to a range the same as the reward function. The author identify the reward hacking causes in PRM and claim that the PURE framework can significantly alleviate this issue. Compared to three base models, the new framework produces higher performance. In particular, it can achieve similar performance with only 30% steps compared to verifiable reward-based methods. This avoids the collapse issue by adopting the cononical summation of rewards approach. The author also demonstrate the cases for reward hacking and analyze training collapse.

**Questions:**

1) Referring to Weakness 2, could you tell what is the main contribution compared to the previous works? Why is the min-form credit assignment still novel in your case?

2) What is the main difference between PURE-VR and other VR methods, and could you describe in more detail? The results in Table 2 do not take into consideration the different moving parts for the base models. Is it possible to include those?

**Ethical Concerns:**

["NO or VERY MINOR ethics concerns only"]

**Final Justification:**

The author answered my question well and eased my main concern regarding the novelty, plus the initial good performance of their methods. Hence, I have raised my score.

**Limitations:**

Yes.

**Paper Formatting Concerns:**

No issue noticed.

**Quality:**

3

**Strengths And Weaknesses:**

## Strengths

1) In general, the paper has an appropriate motivation towards the design of a min-form credit assignment framework, since there requires special curation in RL for language models.

2) The code is simple to implement, as it requires only a transformation of process rewards without additional code changes.

3) PURE achieves more efficiency and demonstrates its advantage in performance compared to the base models. This is well supported by results in Table 2 and Figure 2.

4) The analysis towards reward hacking and collapse is interesting and helpful.

## Weaknesses

1) The author include results and discussion about summation-form credit assignment. However, is min-form based credit assignment the only way to alleviate this? To me, it seems like the problem mainly resides on the lack of data in PRM. It is also common practice in RL to use gradient clip and smaller step size for the optimizer under such scenarios.

2) In line 86 to 88, the author mentioned previous min-value aggratation of process rewards (Lightman et al., 2023; Wang et al., 2023; Zhang et al., 2025), which is also defined by $min(r^p_1, ..., r^p_n)$. The concept is very much similar here, and Eq. (3) and (4) do not include too much novelty either.

3) In Table 2, it is not clear if PURE is actually better than the baselines for all the moving parts.  I assume the appropriate way to compare is such as the base model with VR v.s. PURE with VR, base model with PRM v.s. PURE with PRM etc.

---

> ### Author Rebuttal · Authors · 2025-07-29
>
> Dear Reviewer ydx9,
>
> We sincerely appreciate your valuable and insightful comments as they are extremely helpful for improving our manuscript. We have addressed each comment in detail below.
>
> ---
>
> > **[w1]** The author include results and discussion about summation-form credit assignment. However, is min-form based credit assignment the only way to alleviate this? To me, it seems like the problem mainly resides on the lack of data in PRM. It is also common practice in RL to use gradient clip and smaller step size for the optimizer under such scenarios.
>
> **[A1]** Thank you for raising this question. While min-form credit assignment is not necessarily the only solution for PRM-based RFT training, it currently represents the most practical and effective approach. By applying a simple transformation to process rewards, it converts initially unstable training into a stable process capable of sustained learning over hundreds or thousands of steps.
>
> We respectfully disagree that PRM data limitations are the primary cause of instability in math reasoning RFT, based on three key observations:
>
> 1. First, let’s think about the question of how much data is sufficient. During the RLHF era, InstructGPT utilized 500k preference pairs of data [1], while Claude used around the 1M scale [2]. Our preprocessed PRM800k dataset contains 370k samples with approximately 7 process labels for each sample, totaling 2~3M process labels. In comparison, such scale of PRM data can be considered sufficient. Considering the scaling law of preference labeling [3], if we want to further improve the accuracy of PRM, we would need to label another million-scale dataset - presents prohibitive costs for most institutions. Crucially, our method achieves stable training with existing datasets.
> 2. As Section 5.1 demonstrates, the core issue is not PRM inaccuracy but reward hacking during RL training. In the first two hacking types (Section 5.1), PRMs provide accurate step-wise judgments (Figure 1), but the RL algorithm derives biased advantage values. This problem is fundamentally RL algorithmic rather than data-related.
> 3. We do employ standard RFT stabilization techniques including gradient clipping (norm=1) and reduced learning rates (5e-7 or 1e-6), ensuring minimal per-step updates. Both of these tricks are default configurations in verl, but these measures alone proved insufficient without our credit assignment modification.
>
> [1] Ouyang, Long, et al. "Training language models to follow instructions with human feedback." *Advances in neural information processing systems* 35 (2022): 27730-27744.
>
> [2] Bai, Yuntao, et al. "Training a helpful and harmless assistant with reinforcement learning from human feedback." *arXiv preprint arXiv:2204.05862* (2022).
>
> [3] Hou, Zhenyu, et al. "Does RLHF Scale? Exploring the Impacts From Data, Model, and Method." *arXiv preprint arXiv:2412.06000* (2024).
>
> ---
>
> > **[w2]** In line 86 to 88, the author mentioned previous min-value aggratation of process rewards (Lightman et al., 2023; Wang et al., 2023; Zhang et al., 2025), which is also defined by $\min (r_1^p, \cdots, r_n^p)$. The concept is very much similar here, and Eq. (3) and (4) do not include too much novelty either.
>
> **[A2]** Thank you for highlighting this. Our approach differs from prior work in two critical aspects:
>
> 1. Existing methods apply sample-level min-form during test time, while we define and implement token-level min-form during training time. The most critical issue here is that during testing, the LLM parameters are fixed, meaning that the distribution of LLM’s output does not change over time. Therefore, as long as the PRM adapts to the distribution of the LLM's output, the system will work properly. **However, during training, we must deal with distribution shifts, meaning the distribution of LLM’s output is constantly changing**. Especially in math reasoning, where the length of the entire output continues to increase during training. Additionally, the frequencies of some patterns are activated, while others are suppressed, thus the LLM's output distribution is constantly changing. This places higher demands on the overall algorithm's stability. Simply ensuring the PRM adapt to the initial LLM’s output distribution is insufficient; **the algorithm must continuously ensure that the output/input distributions of the LLM and PRM remain aligned to enable stable training over hundreds or even thousands of steps.** This distinction is similar to the difference between offline (such as native DPO) and online RLHF (such as online iterative DPO [1]). Offline RLHF only needs LLM adapt to a fixed data distribution, while online RLHF requires the entire system to continuously adapt to the evolved data distribution. This requires more sophisticated designs to ensure training stability, but at the same time will also bring better performance [1-2].  Specifically, in our algorithm implementation, we have to carefully decide whether to use the minimum value of future rewards or the minimum value of overall rewards, how to split the steps, whether to introduce a special token or consider all decoded characters containing “\n\n” tokens to split steps, and whether steps after the first error step should be included in PRM training, among other details. These factors impact the overall algorithm stability and ultimately determine whether the entire project can work effectively.
> 2. While prior work uses sample-level aggregation, training requires token-level value functions. Therefore, we need to define an appropriate token-level value function (Eq. (3) and (4)). In the code implementation, to make our algorithm as easy to implement as possible, we propose a novel transformation function, allowing near-theoretical approximation through a single operation that reduces coding complexity.
>
> [1] Xiong, Wei, et al. "Iterative preference learning from human feedback: Bridging theory and practice for rlhf under kl-constraint." *arXiv preprint arXiv:2312.11456* (2023).
>
> [2] Dong, Hanze, et al. "Rlhf workflow: From reward modeling to online rlhf." *arXiv preprint arXiv:2405.07863* (2024).
>
> ---
>
> > **[w3]** In Table 2, it is not clear if PURE is actually better than the baselines for all the moving parts. I assume the appropriate way to compare is such as the base model with VR v.s. PURE with VR, base model with PRM v.s. PURE with PRM etc.
>
> **[A3]** Thank you for pointing this out. We think there exists misunderstanding and would like to clarify as follows:
>
> 1. **The base model with VR is theoretically equivalent to PURE with VR.** Since our framework has made significant modifications to the verl codebase, we must test whether the results can be reproduced under consistent hyperparameter settings. Therefore, we run PURE-VR and compare its performance with SimpleRL-Zoo in Table 2. As mentioned in line 217 of the paper, the settings for PURE with VR and Simple-RL are completely consistent and PURE-VR reproduces SimpleRL-Zoo's performance.
> 2. **The base model with PRM is equal to use sum-form credit assignment with PRM, i.e., PURE-PRM (sum-form) shown in Figure 2.** Therefore, “base model with PRM vs. PURE with PRM” is equivalent to “PURE-PRM (sum-form) vs. PURE-PRM (min-form)”. PURE-PRM (sum-form) even collapses training at the beginning, thus, if we need to report its benchmark scores, we can only use the checkpoint at around step 5, which, as one can expect, won't be good either.
>
> ---
>
> > **[Q1]** Referring to Weakness 2, could you tell what is the main contribution compared to the previous works? Why is the min-form credit assignment still novel in your case?
>
> **[A4]** Thanks for highlighting this. Please refer to **[A2]** for the novelty of the proposed min-form credit assignment method. In addition, the summarized reward hacking cases and the causes of training collapse in Section 5 are also important and novel to the community, which helps the community further understand the corner cases of LLM$\times$RL and thereby research new algorithms.
>
> ---
>
> > **[Q2]** What is the main difference between PURE-VR and other VR methods, and could you describe in more detail? The results in Table 2 do not take into consideration the different moving parts for the base models. Is it possible to include those?
>
> **[A5]** Thanks for pointing this out. Please refer to the detailed response in **[A3]**. Additionally, we would like to supplement that, based on our experience and technical report from ByteDance [1], Nvidia [2], and SkyWork [3], the dynamic and performance of LLM$\times$RL experiments are closely related to the codebase. Even with the same hyperparameters, the results and training dynamics obtained using OpenRLHF and verl will differ a lot. Therefore, after implementing our codebase, it is necessary to conduct point-to-point comparison under the same setting (i.e., only verifiable rewards). As mentioned in line 219 of our paper, the performance of PURE-VR is comparable to SimpleRL-Zoo under exactly same setting and hyperparameters, confirming the reliability of our codebase.
>
> Our modifications to verl under verifiable reward settings include:
>
> - Call reward function before computing log probabillity to support curriculum learning
> - Fix the rare occurrence of reward function hang bug
> - Change the placement of the token-level verifiable rewards from the last of the padded tokens to the last token of un-padded valid tokens to better support multiple reward modes
> - More logged metrics to comprehensively monitor the training dynamics
>
> ​	…
>
> [1] Seed, ByteDance, et al. "Seed1. 5-thinking: Advancing superb reasoning models with reinforcement learning." *arXiv preprint arXiv:2504.13914* (2025).
>
> [2] Chen, Yang, et al. "Acereason-nemotron: Advancing math and code reasoning through reinforcement learning." *arXiv preprint arXiv:2505.16400* (2025).
>
> [3] He, Jujie, et al. "Skywork open reasoner 1 technical report." *arXiv preprint arXiv:2505.22312* (2025).

---

> > ### Comment · Reviewer_ydx9 · 2025-08-04
> > **I raised my score**
> >
> > I thank the author for answering my questions and providing supplements, especially for W2 and Q1. I recognize the contribution of this work towards the community and have raised my score.

---

> > > ### Author Response · Authors · 2025-08-04
> > >
> > > Dear Reviewer ydx9,
> > >
> > > Thank you for your constructive feedback and generous score adjustment. We deeply appreciate your recognition of our contributions. Your encouragement motivates us to continue advancing this line of research.
> > >
> > > Best regards,
> > >
> > > Authors of Paper 2569

---

### Official Review · Reviewer_Pj7M · 2025-06-30

**Clarity:** 3
**Significance:** 3
**Originality:** 3
**Rating:** 4
**Confidence:** 3

**Summary:**

This paper introduces a method designed to reduce reward hacking during reinforcement fine-tuning (RFT) of LLMs. The main idea is to propose replacing traditional summation-based credit assignment with a min-form credit assignment.

**Questions:**

See the comments above.

**Ethical Concerns:**

["NO or VERY MINOR ethics concerns only"]

**Final Justification:**

I have read the author's rebuttal and considered all the raised points. Most of my concerns are addressed.

Regarding the reasonableness of Assumption 1, this assumption can significantly simplify the theoretical derivation. However, if the global bound `ε` from Assumption 1 is large, then the `ε` in the final upper bound will also become large as well. But considering that the experimental performance is very good, I think it empirically suggests that the size of the global bound `ε` is likely relatively small in this setting. Therefore, this is not a major issue; however, if this assumption could be relaxed (or improved), it has the potential to yield a tighter bound.

Hope the authors can further consider this point.

**Limitations:**

Yes

**Quality:**

3

**Strengths And Weaknesses:**

**Pros**:
- The min-form credit assignment is a novel and principled solution that tackles a timely and important problem.

- The paper shows significant improvement across several datasets and models

- Comprehensive analysis and discussion of reward hacking cases, providing actionable insights for further research

**Cons/Questions**:
- I recommend that the authors not use a title like **"sth is all sth need"**, which often exaggerates the facts and shadows the contribution of the paper.

- Theoretical guarantees regarding the min-form credit assignment are not deeply explored.

- The experimental validation is primarily conducted on mathematical reasoning benchmarks and models derived from the Qwen2.5 series,  potentially limiting immediate generalizability to broader domains

- The results are reported based on single-run evaluations, without confidence intervals or multiple seeds. This makes it difficult to assess the statistical robustness and reproducibility of the observed improvements. We cannot tell if the results are statistically significant or generalizable.

- Have the authors conducted any multi-seed experiments? If so, could they report the mean and standard deviation of the performance?

- All experiments are conducted on mathematical reasoning tasks. Can the authors provide evidence or discussion regarding the applicability of PURE to domains such as commonsense reasoning, code generation, or dialogue modeling?

---

> ### Author Rebuttal · Authors · 2025-07-29
>
> Dear Reviewer Pj7M,
>
> We sincerely appreciate your valuable and insightful comments as they are extremely helpful for improving our manuscript. We have addressed each comment in detail in the paragraphs below.
>
> ---
>
> > **[w1]** I recommend that the authors not use a title like **"sth is all sth need"**, which often exaggerates the facts and shadows the contribution.
>
> **[A1]** We appreciate the reviewer's suggestion regarding title clarity. We will revise the title to **"Stop Summation: Min-Form Credit Assignment Suffices for Process Reward Models in Mathematical Reasoning"** to more precisely reflect our contribution's scope. This maintains our core insight that minimal-form credit assignment can effectively replace summation in process-based reward modeling.
>
> ---
>
> > **[w2]** Theoretical guarantees regarding the min-form credit assignment are not deeply explored.
>
> **[A2]** Thanks for highlighting this. We derive the following theorem to theoretically analyze our algorithm:
>
> **Assumption:**
>
> We first establish 2 reasonable assumptions:
>
> * **Assumption 1 (Bounded Process Reward Error):** The Process Reward Model, $R^p$, produces a reward estimate $r_t^p = R^p(s_t, a_t)$ for each reasoning step. It is assumed that there exists a "true" or optimal reward $r_t^*$ for that step, and the estimation error is uniformly bounded by a constant $\epsilon \ge 0$. For any step $t$, it holds that $|r_t^p - r_t^\*| \le \epsilon$.
>
> * **Assumption 2 (Bounded Rewards):** The true rewards, and consequently the estimated rewards, are bounded. There exists a maximum possible reward $R_{max}$ such that $|r_t^*| \le R_{max}$ and $|r_t^p| \le R_{max} + \epsilon$ for all steps $t$.
>
> **Theorem (Q-value Estimation Error Bound Comparison):**
> Under Assumptions 1 and 2, for any state-action pair $(s_t, a_t)$ and a trajectory $\tau$ with $n$ reasoning steps:
>
> 1.  **Sum-form Error Bound:** The estimation error for the summation-form Q-value, is bounded by:
>     $$|Q_{sum}^{\pi}(s_t, a_t) - Q_{sum}^{*}(s_t, a_t)| \le \epsilon \sum_{i=t}^{n} \gamma^{i-t}$$
>
>     For an infinite horizon ($n \to \infty$), this bound becomes:
>     $$|Q_{sum}^{\pi}(s_t, a_t) - Q_{sum}^{*}(s_t, a_t)| \le \frac{\epsilon}{1-\gamma}$$
>
>     This shows that the error bound is amplified by a factor dependent on the horizon length and the discount factor $\gamma$, accumulating the error over steps.
>
> 2.  **Min-form Error Bound:** The estimation error for the min-form Q-value, derived from Equations (3) and (4) of the paper, is bounded by:
>     $$|Q_{min}^{\pi}(s_t, a_t) - Q_{min}^{*}(s_t, a_t)| \le \epsilon$$
>
>     This shows that the error bound is strictly limited by the single-step reward estimation error $\epsilon$ and **does not accumulate** with the number of steps.
>
> **Proof Part 1: Proof for the Sum-form Error Bound**
>
> 1.  We begin with the definition of the absolute error of the Q-value, using the canonical formulation from the paper's Equation (1). $Q_{sum}^{\pi}$ denotes the value calculated with the estimated rewards from the PRM ($r^p$), and $Q_{sum}^{\*}$ denotes the true value (calculated with $r^*$).
>
>     $$ |Q_{sum}^{\pi}(s_t, a_t) - Q_{sum}^{\*}(s_t, a_t)| = \left| E_{\tau \sim \pi} \left[ \sum_{i=t}^{n} \gamma^{i-t} r_i^p \right] - E_{\tau \sim \pi} \left[ \sum_{i=t}^{n} \gamma^{i-t} r_i^\* \right] \right| = \left| E_{\tau \sim \pi} \left[ \sum_{i=t}^{n} \gamma^{i-t} (r_i^p - r_i^\*) \right] \right| $$
>
>     $$\le E_{\tau \sim \pi} \left[ \left| \sum_{i=t}^{n} \gamma^{i-t} (r_i^p - r_i^\*) \right| \right] \le E_{\tau \sim \pi} \left[ \sum_{i=t}^{n} \left| \gamma^{i-t} (r_i^p - r_i^\*) \right| \right] = E_{\tau \sim \pi} \left[ \sum_{i=t}^{n} \gamma^{i-t} |r_i^p - r_i^\*| \right]$$
>
> 2. Now, we apply **Assumption 1**, which states that the reward error at each step is bounded by $\epsilon$, i.e., $|r_i^p - r_i^*| \le \epsilon$:
>
>    $$|Q_{sum}^{\pi}(s_t, a_t) - Q_{sum}^{\*}(s_t, a_t)| \le E_{\tau \sim \pi} \left[ \sum_{i=t}^{n} \gamma^{i-t} \epsilon \right] = \epsilon \sum_{i=t}^{n} \gamma^{i-t}$$
>
>    For the infinite horizon case, the geometric series converges to $\frac{1}{1-\gamma}$, which completes the first part of the proof.
>
> **Proof Part 2: Proof for the Min-form Error Bound**
>
> 1.  We start similarly with the definition of the absolute Q-value error for the min-form, based on Equations (3) and (4) from the paper. The return $G(s_t, a_t|\tau)$ is the value being expected.
>
>     $$|Q_{min}^{\pi}(s_t, a_t) - Q_{min}^{\*}(s_t, a_t)| = \left| E_{\tau \sim \pi} [G(s_t, a_t|\tau)] - E_{\tau \sim \pi} [G^\*(s_t, a_t|\tau)] \right| \le E_{\tau \sim \pi} \left[ |G(s_t, a_t|\tau) - G^\*(s_t, a_t|\tau)| \right]$$
>
> 3.  Now we analyze the return term inside the expectation. According to Equation (3), for a trajectory $\tau$ and a step $t$, the return is defined as the minimum of future rewards.
>     * $G(s_t, a_t|\tau) = \min(r_t^p, r_{t+1}^p, \dots, r_n^p)$
>     * $G^\*(s_t, a_t|\tau) = \min(r_t^\*, r_{t+1}^\*, \dots, r_n^\*)$
>
> 4.  We use a key property of the minimum function: $| \min(\mathbf{a}) - \min(\mathbf{b}) | \le \max_i |a_i - b_i|$. Applying this property:
>
>     $$|G(s_t, a_t|\tau) - G^\*(s_t, a_t|\tau)| = |\min_{i=t \dots n}(r_i^p) - \min_{i=t \dots n}(r_i^\*)| \le \max_{i=t \dots n} |r_i^p - r_i^\*| \le \epsilon$$
>
> 4. Substitute this result back into the expectation inequality from step 1:
>
>    $$|Q_{min}^{\pi}(s_t, a_t) - Q_{min}^{\*}(s_t, a_t)| = E_{\tau \sim \pi} [|G(s_t, a_t|\tau) - G^\*(s_t, a_t|\tau)|] \le E_{\tau \sim \pi} [\epsilon] = \epsilon$$
>
>    This completes the proof. It is proven that the error in the min-form does not accumulate and is directly bounded by the maximum single-step error of the reward model.
>
> **Furthermore, we also derive upper bounds for both the advantage function error and the update gradient error.** Due to rebuttal's word limit, we will add these two theorems in the appendix.
>
> ---
>
> > **[w3]** The experimental validation is primarily conducted on mathematical reasoning benchmarks and models derived from the Qwen2.5 series, potentially limiting immediate generalizability to broader domains
>
> **[A3]** Thank you for this observation. For experiments on other models, we conduct additional experiments using Llama-3.1-8B with PURE-PRM+VR, compared against SimpleRL-Zoo (theoretically equivalent to PURE-VR). Results confirm that our algorithm maintains performance advantages on the Llama-3.1 architecture.
>
> | model        | method         | MATH-500 | Minerva Math | Olympiad Bench | AIME2024 | AMC23 | AVG.  |
> | ------------ | -------------- | -------- | ------------ | -------------- | -------- | ----- | ----- |
> | Llama-3.1-8B | - (Base model) | 13.6     | 4.8          | 3.1            | 0.0      | 2.5   | 4.8   |
> | Llama-3.1-8B | SimpleRL-Zoo   | 23.0     | 9.6          | 5.3            | 0        | 10.0  | 9.58  |
> | Llama-3.1-8B | PURE-PRM+VR    | 26.0     | 10.3         | 5.5            | 3.3      | 15.0  | 12.02 |
>
> For experiments beyond mathematical reasoning, please refer to **[A6]** for our detailed response.
>
> ---
>
> > **[w4]** The results are reported based on single-run evaluations, without confidence intervals or multiple seeds. This makes it difficult to assess the statistical robustness and reproducibility of the observed improvements. We cannot tell if the results are statistically significant or generalizable.
>
> **[A4]** Thank you for emphasizing statistical rigor. We conduct three additional runs with Qwen2.5-Math-7B using three different random seeds. The table below shows mean performance and standard deviation across four runs, demonstrating stable average results with only 0.84 standard deviation across multiple benchmarks.
>
> | random seed | MATH-500 | Minerva Math | Olympiad Bench | AIME2024 | AMC23 | AVG. |
> | ----------- | -------- | ------------ | -------------- | -------- | ----- | ---- |
> | 42          | 82.6     | 37.1         | 44.1           | 20       | 82.5  | 53.3 |
> | 12345       | 82.2     | 37.1         | 44             | 23.3     | 75    | 52.3 |
> | 23451       | 82.6     | 40.4         | 43.3           | 23.3     | 70    | 51.9 |
> | 34512       | 82       | 37.5         | 44.7           | 20       | 72.5  | 51.3 |
> | **mean**    | 82.35    | 38.025       | 44.025         | 21.65    | 75    | 52.2 |
> | **std**     | 0.30     | 1.59         | 0.57           | 1.91     | 5.40  | 0.84 |
>
> ---
>
> > **[w5]** Have the authors conducted any multi-seed experiments? If so, could they report the mean and standard deviation of the performance?
>
> **[A5]** Thank you for your inquiry. Please refer to **[A4]** for detailed results from our multi-seed experiments.
>
> ---
>
> > **[w6]** All experiments are conducted on mathematical reasoning tasks. Can the authors provide evidence or discussion regarding the applicability of PURE to domains such as commonsense reasoning, code generation, or dialogue modeling?
>
> **[A6]** We appreciate your interest in broader applications. We are actively exploring PURE's extension to general reasoning domains (math, code, science, logic), which presents significant challenges. The primary limitation is not our RL algorithm but rather scoring inaccuracies from PRMs. Since the PRM800k dataset (the largest open-source PRM training data) lacks corner-case responses and diverse domain data, its out-of-distribution characteristics lead to unreliable step-level scoring.
>
> To address this, we are:
>
> 1. Manually labeling 100k process labels for diverse domains
> 2. Allocating ~1,000 H200 GPU hours for dedicated training
>
> We are systematically addressing these challenges one by one. Currently, we have experimented with Qwen2.5-7B in the mixed field of math and science, **increasing the GPQA Diamond (a science benchmark) score from 38 to 53 using PURE-PRM+VR**, compared with 50 of PURE-VR. This result shows that there is no fundamental difference between different domains in terms of algorithms. It is merely a matter of different domain data ratios. We plan to release this new project in the near future.

---

### Official Review · Reviewer_GpAR · 2025-07-03

**Clarity:** 3
**Significance:** 3
**Originality:** 3
**Rating:** 4
**Confidence:** 4

**Summary:**

The paper introduces process supervised reinforcement learning, a reinforcement fine-tuning framework designed to stabilize training of large language models using dense, step-level feedback from process reward models. The authors identify a key issue in prior work: summation-form credit assignment often causes LLMs to exploit high-reward steps, leading to reward hacking and training collapse. This method replaces the summation with a novel min-form credit assignment, which focuses on the worst-performing step in a reasoning chain. Experiments across multiple benchmarks show that PURE achieves comparable reasoning performance to traditional methods with higher training efficiency.

**Questions:**

1.  It is recommended to use argmax instead of maximize in Eq.(2) to keep the consistency of symbol usage.
2. Some queries and improvement suggestions for Fig.2:
a)  Why are the accuracy curves recorded on the 1.5B model, not 7B?
b)  It is suggested to explain the reason that some curves are truncated in the caption (e.g., early stop caused by training divergence).

**Ethical Concerns:**

["NO or VERY MINOR ethics concerns only"]

**Final Justification:**

I have read the author's rebuttal and considered all the other reviewers' raised points. Most of my concerns are addressed, so I keep my acceptance rating.

**Limitations:**

yes

**Quality:**

3

**Strengths And Weaknesses:**

Strengths:
1.	By replacing the traditional summation-based credit assignment with a min-form approach, the proposed PURE framework significantly reduces reward hacking.
2.	The framework flexibly incorporates both dense and sparse rewards, and shows that even a small fraction of ground-truth signals can substantially enhance final performance and robustness.
3.	The min-form credit assignment in PURE is implemented by simply transforming process rewards without altering the reinforcement learning pipeline or core code structure

Weaknesses:
1. In Fig.3, it seems that the results in the sub-graph (top center) after approximately step 370 are truncated. Please consider log-scaling on the y-axis if this is not caused by training interruption.
2. Please provide more accurate curves (like Fig.3(b)) on different datasets and different parameter models in the  Appendix, to better analyze the stability of the proposed method in the whole training cycle.
3.  A minor error in page 8, line 288: duplicated periods.

---

> ### Author Rebuttal · Authors · 2025-07-29
>
> Dear Reviewer GpAR,
>
> We sincerely appreciate your valuable and insightful comments as they are extremely helpful for improving our manuscript. We have addressed each comment in detail below.
>
> ---
>
> > **[w1]** In Fig.3, it seems that the results in the sub-graph (top center) after approximately step 370 are truncated. Please consider log-scaling on the y-axis if this is not caused by training interruption.
>
> **[A1]** Thank you for highlighting this observation. At approximately step 370, the clip ratio for both correct and incorrect responses rises sharply to around 0.85. This upward trend aligns with concurrent increases in response length (shown in Figure 3, top left). We agree that log-scaling the y-axis would improve visualization. Thank you very much for your suggestion! Due to NeurIPS rebuttal policies, we cannot include images this year but will update this subfigure in the final manuscript.
>
> ---
>
> > **[w2]** Please provide more accurate curves (like Fig.3(b)) on different datasets and different parameter models in the Appendix, to better analyze the stability of the proposed method in the whole training cycle.
>
> **[A2]** Thank you for this suggestion. This is indeed crucial for the analysis of RL algorithms. For Qwen2.5-Math series, the accuracy on MATH-500 rises steadily within 500 training steps for PURE-VR, PURE-PRM+VR, and PURE-PRM (until its collapse). For Qwen2.5-7B:
>
> - PURE-PRM: accuracy rises steadily before collapsing.
> - PURE-PRM+VR: accuracy rises for 500 steps, converges, then fluctuates around the converged value.
> - PURE-VR: accuracy rises steadily through 1000 steps, reaching PURE-PRM+VR’s convergence value at step 1600.
>
> Per current NeurIPS rebuttal constraints (no links and images), we cannot display them here. We will add full training dynamic figures to the appendix in the final manuscript.
>
> ---
>
> > **[w3]** A minor error in page 8, line 288: duplicated periods.
>
> **[A3]** Thank you for catching this typo. We will correct the duplicated periods and carefully review the manuscript for similar errors.
>
> ---
>
> > **[Q1]** It is recommended to use argmax instead of maximize in Eq.(2) to keep the consistency of symbol usage.
>
> **[A4]** Thank you for this suggestion. We will update Eq.(2) accordingly. Once again, we sincerely appreciate your meticulous review, which greatly improves the quality of our manuscript!
>
> ---
>
> > **[Q2]** Some queries and improvement suggestions for Fig.2: a) Why are the accuracy curves recorded on the 1.5B model, not 7B? b) It is suggested to explain the reason that some curves are truncated in the caption (e.g., early stop caused by training divergence).
>
> **[A5]** Thank you for pointing this out.
>
> **a)** In order to show as much of the training dynamics of each model as possible within page limits, we assigned Figure 2a to Qwen2.5-Math-7B, Figure 2b to Qwen2.5-Math-1.5B, and Figure 3 to Qwen2.5-7B. Accuracy curves for 7B models are detailed in **[A2]**.
>
> **b)** We appreciate this suggestion. That would indeed be more reasonable! The x-axis of some curves in Figure 2 are truncated due to training collapse. The y-axis of the clip ratio in Figure 3 (top center) is truncated to highlight early-stage changes near $y=0$ (will use log-scaling later). The curve of $T=1$ in Figure 7 is truncated due to training convergence.

---

### Official Review · Reviewer_GZwS · 2025-07-15

**Clarity:** 2
**Significance:** 2
**Originality:** 2
**Rating:** 4
**Confidence:** 2

**Summary:**

The paper proposes PURE, a min-form credit assignment that formulates the value function as the minimum of future rewards. This method significantly alleviates reward hacking by limiting the value function range and distributing advantages more reasonably. Authors test PURE on 3 different base models and demonstrate superiority against the sum-form credit asssignment which they claim collapses training. The paper discusses observed reward hacking cases and also provide an analysis on the training collapse.

**Questions:**

1. How does the min-form credit assignment behave when the future rewards in a reasoning trace are consistently very low or zero for several steps, and does this lead to specific challenges or biases in the learning process?

2. How would the PURE framework perform in real-world applications where PRMs might be more noisy or less reliable in their reward assignments compared to the experimental setup?

3. Beyond reasoning tasks, could the min-form credit assignment principle be effectively applied to other fine-tuning scenarios (e.g., instruction following) where process-based feedback might be available?

**Ethical Concerns:**

["NO or VERY MINOR ethics concerns only"]

**Limitations:**

Please refer to weaknesses section.

**Quality:**

3

**Strengths And Weaknesses:**

Strengths:

1. The paper is nicely motivated and clearly discusses where and how the baseline approaches struggle. The experiments are diverse and satisfactory to convince the claims made in introduction.

2. The min-form assignment captures the limitation of process rewards in a simple yet effective manner. The implementation is simple, and aligned with the commonly used token-level PPO loss. The effectiveness of this simple approach is captured well in both qualitative and quantitative results.


Weaknesses:

1. While the paper clearly defines reward hacking in the context of summation-form credit assignment (exploiting high-reward steps), it doesn't explicitly discuss whether the min-form approach might introduce other, subtler forms of exploitation or biases in the learning process, particularly if the minimum reward is consistently low across many steps.

2. The paper demonstrates success on reasoning tasks, but the complexity and length of these reasoning chains are not fully detailed. It would be beneficial to understand how the min-form approach scales to extremely long and intricate reasoning processes where maintaining a minimum reward across all steps could be challenging.

---

> ### Author Rebuttal · Authors · 2025-07-29
>
> Dear Reviewer GZwS,
>
> We sincerely appreciate your valuable and insightful comments as they are extremely helpful for improving our manuscript. We have addressed each comment in detail in the paragraphs below.
>
> ---
>
> > **[w1]** While the paper clearly defines reward hacking in the context of summation-form credit assignment (exploiting high-reward steps), it doesn't explicitly discuss whether the min-form approach might introduce other, subtler forms of exploitation or biases in the learning process, particularly if the minimum reward is consistently low across many steps.
>
> **[A1]** Thank you for raising this important point. In Section 5.1, we document all observed cases of reward hacking across our experiments. While summation-form credit assignment exhibited all three hacking cases, **min-form credit assignment only manifested the third type** – identical to the case at step 361 in Section 5.2. Specifically:
>
> > *The model only generates meaningless responses (e.g., "Thank you", or empty outputs “”).*
>
> This occurs because discriminative PRMs assign high rewards to such irrelated response steps, not realizing that no further content follows – a fundamental limitation of current PRM architectures (i.e., the causal attention mask). Crucially, **this behavior is independent of the RL algorithm** and affects *both* min-form and sum-form credit assignment methods.
>
> To mitigate this, we use verifiable rewards as auxiliary guidance with proper reward coefficients:
>
> - Effective configuration: 1.0 * verifiable rewards + 1.0 * process rewards, used in main experiments (Sec 4.3)
> - Failure case: 1.0 * verifiable rewards + 2.0 * process rewards → collapse (Sec 5.2)
>
> Regarding your concern about persistent low minimum rewards, we provide a detailed analysis in **[A3]** below.
>
> ---
>
> > **[w2]** The paper demonstrates success on reasoning tasks, but the complexity and length of these reasoning chains are not fully detailed. It would be beneficial to understand how the min-form approach scales to extremely long and intricate reasoning processes where maintaining a minimum reward across all steps could be challenging.
>
> **[A2]** Thank you for highlighting this. To understand the complexity and length of the reasoning chains, we logged the response length and frequnencies of reflection patterns (e.g., “wait,”, “recheck”, “retry”, “alternatively,”, “let's verify”, and so on). For Qwen2.5-Math-7B, the average response length increases monotonically from 750 tokens to around 2000 tokens over 1000 training steps. The average reflection frequency increases from 0.3 to 0.9 in the first 700 steps and fluctuates around 0.9 for the final 300 steps. Following NeurIPS rebuttal policies prohibiting links and images this year, we will include full training dynamics figures in the appendix. These cover nearly 20 metrics spanning sample-level (verifiable/process rewards, returns, advantages), group-level (pass@n, avg@n), benchmark-level (MATH-500, AIME24), pattern-related (repetition score, reflection frequency, response length), stability-related (actor grad norm, response clip ratio), and system metrics (iteration/generation time, step count), etc.
>
> Regarding ultra-long reasoning processes (such as average response length > 10k), this remains challenging for both PRM-based and VR-based methods, representing an ongoing research focus for our team. The primary obstacle is not the RL algorithm itself but rather scoring inaccuracies from PRMs. Since the PRM800k dataset – the largest open-source PRM training data – lacks such long responses, its out-of-distribution characteristics lead to unreliable step-level scoring for ultra-long responses. We are actively addressing this through manual labeling of process rewards for 100k responses (generated by deepseek-R1-0120) and dedicating approximately 1,000 H200 GPU hours for training. **Importantly, our current observations indicate that for ultra-long reasoning, min-form credit assignment still demonstrates greater stability than summation-form.** Specifically, as step counts increase, the summation-form method consistently exhibits reward hacking at certain steps, as if its training had “ten thousand” ways to fail.
>
> ---
>
> > **[Q1]** How does the min-form credit assignment behave when the future rewards in a reasoning trace are consistently very low or zero for several steps, and does this lead to specific challenges or biases in the learning process?
>
> **[A3]** Thank you for raising this concern. Consistently low-reward steps do not inherently disrupt training because **the advantage function - not the raw rewards - determines the magnitude and direction of policy updates** (see the PPO loss). For advantage estimators like RLOO and GRPO, the advantage is computed as the reward minus a baseline that depends on other samples within the group. Thus, even if certain steps receive low rewards, their advantage values won't necessarily approach zero unless all samples in the group share identical rewards (i.e., all correct or all incorrect). In such cases, standard practice involves removing these grouped samples (i.e., dynamic sampling proposed by DAPO [1]). Therefore, consistently low rewards across multiple steps do not impede the overall training process.
>
> [1] Yu, Qiying, et al. "Dapo: An open-source llm reinforcement learning system at scale." *arXiv preprint arXiv:2503.14476* (2025).
>
> ---
>
> > **[Q2]** How would the PURE framework perform in real-world applications where PRMs might be more noisy or less reliable in their reward assignments compared to the experimental setup?
>
> **[A4]** Thank you for highlighting this practical consideration. We suggest three approaches when PRM rewards may be noisy or unreliable:
>
> 1. Robust learning [1]: Since training with noisy samples remains an active research area in deep learning, techniques like label smoothing [2] and t-CE loss [3] could be applied directly to PRMs to enhance their noise tolerance.
> 2. Generative PRMs [4]: As discussed in Section 5.1, generative PRMs could leverage the inherent knowledge of foundation models to provide more robust judgments compared to discriminative PRMs.
> 3. Larger Group Sizes: Increasing group size improves baseline stability in advantage estimation. Even if PRMs misassign rewards to some samples, this approach minimizes the impact on policy updates for correctly rewarded samples.
>
> As PRM robustness falls beyond this paper's scope, we defer empirical validation of these approaches to future work.
>
> [1] Song, Hwanjun, et al. "Learning from noisy labels with deep neural networks: A survey." IEEE transactions on neural networks and learning systems 34.11 (2022): 8135-8153.
>
> [2] Wei, Jiaheng, et al. "To smooth or not? when label smoothing meets noisy labels." arXiv preprint arXiv:2106.04149 (2021).
>
> [3] Feng, Lei, et al. "Can cross entropy loss be robust to label noise?." Proceedings of the twenty-ninth international conference on international joint conferences on artificial intelligence. 2021.
>
> [4] Liu, Zijun, et al. "Inference-time scaling for generalist reward modeling." *arXiv preprint arXiv:2504.02495* (2025).
>
> ---
>
> > **[Q3]** Beyond reasoning tasks, could the min-form credit assignment principle be effectively applied to other fine-tuning scenarios (e.g., instruction following) where process-based feedback might be available?
>
> **[A5]** Thanks for this valuable suggestions. We agree that instruction following and similar scenarios present promising applications. As we mentioned in **[A2]**, we are currently exploring towards general reasoning (mathematics, coding, science, and logic), which require overcoming significant computational and data challenges. The primary constraints are GPU resources and the scarcity of process-supervised training data in broader domains. We are systematically addressing these limitations and plan to release new projects extending this framework in the near future.

---

> ### Author Response · Authors · 2025-08-06
>
> Dear Reviewer GZwS,
>
> We sincerely appreciate your time and thoughtful review of our paper. Your feedback has been invaluable in improving our work.
>
> We've carefully addressed all your comments in our rebuttal, including:
>
> 1. Clarification of reward hacking cases in min-form credit assignment
>
> 2. Our ongoing work addressing long-reasoning challenges and generalization
>
> 3. Detailed analysis of algorithm behavior with consistently low rewards
>
> 4. Three practical approaches to handle PRM noise
>
> Please let us know if you need any further clarification or additional questions.
>
> Best regards,
>
> Authors of Paper 2569

---

### Decision · Program_Chairs · 2025-09-17

**Decision:**

Accept (poster)

**Comment:**

This paper addresses reward hacking issues in Process Reward Models (PRMs) when fine-tuning large language models on mathematical reasoning tasks. The authors propose PURE (Process sUpervised Reinforcement lEarning), which replaces the summation-form credit assignment with min-form credit assignment. The paper conducts extensive experiments to demonstrate empirically that their approach successfully tackles the issue for reward hacking.

The reviewers agree that the paper is well-motivated and tackles an important problem (Reviewer GZwS, Pj7M, ydx9). The proposed solution is simple yet effective (Reviewer GZwS, GpAR, ydx9) and the empirical results support their claim (Reviewer GZwS, Pj7M, ydx9).

A major concern is the novelty compared to previous works which also apply min-aggregation (Reviewer ydx9). The authors effectively clarified a crucial distinction: existing methods apply min-aggregation at test-time with fixed model parameters, while their approach operates during training-time where the model distribution continuously shifts.

Reviewer Pj7M raises several concerns including (1) Lack of theoretical guarantees; (2) Single-run evaluations without statistical validation; (3) Limited scope to mathematical reasoning tasks. In response, the authors provide (1) comprehensive analysis comparing error bounds; (2) Multi-seed experimental results; (3) Additional validation on Llama-3.1-8B architecture; (4) Discussion of ongoing work extending to broader domains.

Reviewer GZwS questions about the performance on ultra-long reasoning chains, suggesting that maintaining minimum reward across many steps could be challenging. The authors acknowledge this as an ongoing research challenge affecting both PRM-based and VR-based methods.

Given the strengths mentioned above, as well as the fact that most concerns are addressed in the rebuttal phase, **I lean towards acceptance**.